# Referrals for physical therapy for osteoarthritis during the COVID-19 pandemic: A retrospective analysis

**Manmeet Kaur**[1]*, **Daniel Black**[2], **Jeffrey Fine**[3], **Barton L. Wise**[4]

**1** Department of Internal Medicine, University of California, Davis, Sacramento, CA, United States of America, **2** Division of Clinical Affairs Division of Quality and Safety, Department of Medicine, University of California, Davis, Sacramento, CA, United States of America, **3** Department of Public Health Sciences, University of California, Davis, Sacramento, CA, United States of America, **4** Division of Rheumatology, Department of Orthopaedic Surgery, Department of Medicine, University of California, Davis, Sacramento, CA, United States of America

\* mmkaur@ucdavis.edu

## Abstract

**Data Availability Statement:** All relevant data are within the manuscript and its Supporting Information files. In addition, provided the raw data in the supporting documents (S2 File 2).

### Background

Osteoarthritis (OA) is a leading cause of musculoskeletal pain and disability among Americans. Physical therapy (PT) is recommended per the 2019 ACR /Arthritis Foundation Guideline for Treatment of OA of the Hand, Hip, and Knee. During COVID-19, access to healthcare has been altered in a variety of clinical settings, with the pandemic creating delays in healthcare, with an unknown impact on access to PT care for OA.

### Objectives

We sought to determine whether referrals to PT for OA were reduced in 2020 during the COVID-19 pandemic compared to 2019.

### Methods

A retrospective analysis was done of 3586 PT referrals placed by the University of California, Davis for 206 OA ICD-10 codes from January to November 2019 and from January to November 2020. The numbers of PT referrals per month of each year were compared using both descriptive statistics and Poisson Regression analysis.

### Results

A total of 1972 PT referrals for OA were placed from January to November 2019. Only 1614 referrals for OA were placed from January to November 2020, representing a significant decrease ($p = 0.001$). Month-by-month analysis of 2020 compared to 2019 revealed statistically significant drops in PT referrals for OA in April ($p = 0.001$), May ($p = 0.001$), and August ($p = 0.001$).

**Funding:** Mr. Jeffrey Fine, MPH, who is under an NIH Grant and receives a salary through this grant (UL1 TR001860). As a result, his work is supported by the National Center for Advancing Translational Sciences, National Institutes of Health, through grant number UL1 TR001860. The content is solely the responsibility of the authors and does not necessarily represent the official views of the NIH. This grant had no role in study design, data collection and analysis, decision to publish, or preparation of the manuscript. Overall, the authors received no specific funding for this study.

**Competing interests:** The authors have declared that no competing interests exist.

## Conclusions

These findings reveal a significant reduction in the number of referrals for PT for OA placed in 2020 during the first year of the COVID-19 pandemic. These reductions were particularly evident in the months following state-mandated actions and closures. Factors associated with this outcome may include decreased access to primary care providers, perceptions of PT availability by health care providers, decreased mobility limiting access to both clinic and PT appointments, and/or willingness to engage in PT by patients during the pandemic.

## Introduction

Osteoarthritis (OA) was diagnosed in 32.5 million adults in the U. S. between 2008 to 2014 and is a leading cause of musculoskeletal pain and disability [1, 2]. The goals of care for patients with OA include efforts to reduce pain and improve function [1]. Multi-modal treatment options include pharmacotherapy, physical therapy (PT), and surgery.

PT was recognized and included as recommended treatment by the 2019 American College of Rheumatology/Arthritis Foundation Guideline for the Management of OA of the Hand, Hip, and Knee [3] and can involve a wide range of therapeutic modalities. A review by Brakke et al. [1] detailed how strength training, as well as aquatic, balance, and perturbation therapy, can reduce pain and improve functionality. A similar study performed by Deyle et al. [4] that evaluated the impact of PT on reducing pain and stiffness and improving function in a cohort of adult patients with chronic knee OA reported that PT (both manual therapy and active exercise) resulted in improved scores on the Western Ontario and McMaster Universities Osteoarthritis (WOMAC OA) Index at one year compared to glucocorticoid injections.

While the benefits of PT for the treatment of OA have been recognized, significant patient and physician-associated barriers to engagement remain. For example, one study reported fatalistic attitudes about activity and OA among both patients and general practitioners. Both were reported to be "ambivalent with regards to activity/exercise, recognizing its potential therapeutic usefulness but identifying it at the same time as a cause of knee OA onset or deterioration" [5]. The concern that physical activity might accelerate the progression of the disease was identified as a source of hesitation by both the provider and patient. This concern was compounded by confusion regarding the type and amount of activity that might be necessary to obtain therapeutic benefits without resulting in further joint damage. Concerns regarding ineffective analgesia were identified as additional barriers, as physicians reported difficulties in achieving pain control that would be sufficient to permit OA patients to engage in PT. Patient motivation also influenced compliance with PT, most notably among those patients who had been sedentary throughout life [5].

With the limitations imposed by the Coronavirus Disease-2019 (COVID-19) pandemic, there are reasons to suspect that barriers to PT in OA patients may have increased. Responses to the pandemic have included office closures, reductions in non-urgent care (e.g., cancellation of elective surgery), increased use of telemedicine, and increased fear of healthcare facilities [6]. The pandemic has created a backlog of necessary care and economic instability among both patients and health systems as access to care has been drastically reduced [6]. These pandemic-associated factors may also have an impact on the frequency of patient referrals for PT as well as on patient access to and/or willingness to engage in treatment.

Access to PT during the pandemic may have been even more limited among specific cohorts of OA patients, most notably those of advanced age and/or who belong to

economically marginalized populations. While OA has a particular impact on the elderly, it also disproportionately affects ethnic minorities including those who are Hispanic and non-Hispanic Black. One study revealed that a higher proportion of Hispanic and non-Hispanic Black adults age 45 and older are diagnosed with OA than are non-Hispanic white adults [2].

This study aimed to evaluate the frequency of referrals for PT during 2020, the first year of the COVID-19 pandemic, among patients diagnosed with OA. We hypothesized that there were fewer referrals for PT in 2020 compared to 2019.

## Materials and methods

The study was reviewed and approved by the Institutional Review Board of the University of California (UC) Davis Hospital (Approval # 1713780–1). The study was deemed exempt given no interaction with human subjects and there was no consent needed as the data through de-identified records. The study was a retrospective analysis of the electronic medical record (EMR) database and archived data conducted at UC Davis Hospital in Sacramento, California, USA, and affiliated outpatient centers.

The study included records for both male and female patients between 18–100 years of age who had been assigned one or more of the 206 applicable International Classification of Diseases (ICD)-10 diagnosis codes for OA (see S1 File) with links to PT referrals placed by staff at the UC Davis Hospital and outpatient centers from January 1, 2018, to November 30, 2020. Patients diagnosed with inflammatory arthropathies were excluded. All referrals for PT were included. We did not differentiate between PT referrals that were pending *versus* those that had been completed. Data abstracted from the EMRs included de-identified referral information, specific OA diagnoses based on the ICD-10, and PT referrals made per month and per year during the aforementioned time range.

The EPIC EMR system (Verona, Wisconsin, USA) was used to identify potential study participants who matched the defined demographic criteria. Individual EMR records resulting from our data query were reviewed as applicable. All patient data were de-identified. Computations were performed for each month from January to November of the years 2019 and 2020 to generate summary statistics. Referral and quantitative data for 2018 were reviewed and analyzed for each quarter to identify trends and to determine whether results from 2019 could serve as appropriate controls. Quantitative data were analyzed via the Poisson Means Test with guidance provided by the UC Davis Clinical and Translational Science Center (CTSC) for determining appropriate statistical analyses. A two-factor Poisson Regression was performed to generate a detailed comparison of the 2019 and 2020 data and to evaluate interactions between months and years. A follow-up Šidák step-down p-value adjustment was used to correct for type I error. All statistical analyses were performed with a two-sided alpha of 0.05. Descriptive statistics performed with Microsoft Excel were used to report outcomes. Poisson Regression was performed with SAS® software for Windows® version 9.4 (SAS Institute Inc., Cary, NC).

## Results

Overall PT referral trends (years 2018–2020) for patients diagnosed with OA are shown in Fig 1. We identified 583 PT referrals in the first quarter of 2020 and >500 for each quarter in 2019. We then performed a series of analyses to determine whether 2019 could be used as an appropriate month-to-month control for PT referrals placed in 2020. Initially, we compared PT referral counts in 2018 to those in 2019 and 2020 (shown in Table 1). Our analysis revealed significant reductions in the number of PT referrals in quarters 2 and 3 of 2018 compared to 2019. Given this variation and that all four quarters of 2019 were most similar to the findings

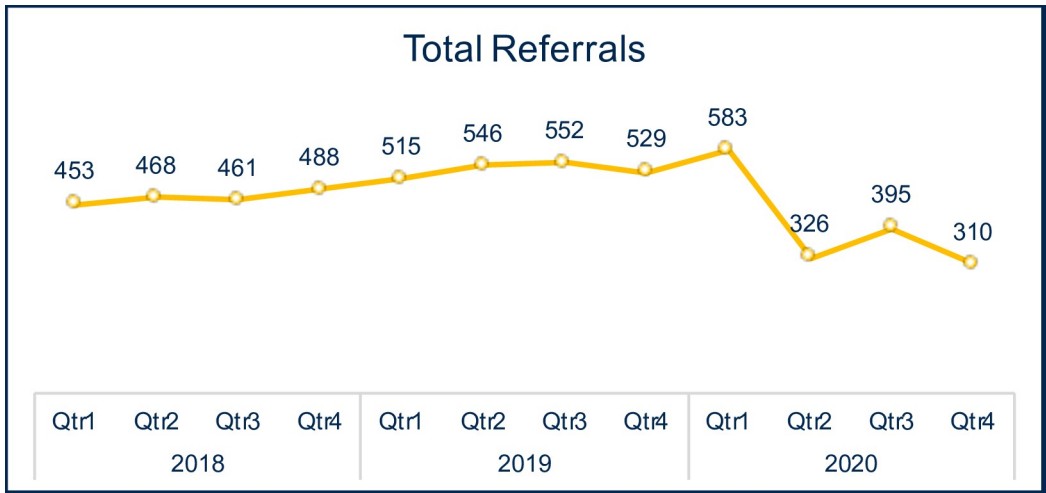

**Fig 1. Total PT referrals per quarter per year.** The yellow line depicts the referral trend from 2018 to 2020, with data with count information per each quarter in each year, showing 2018 and 2019 were fairly similar. However, 2019 had referrals per quarter in the >500, most similar to Quarter 1 of 2020 at 583. Overall, 2020 had reduced referrals quarter 2 onward relative to 2018 and 2019.

in pre-COVID 2020 Quarter 1, 2019 was used for the statistical month-to-month analysis. Collectively, our findings did reveal an overall reduction in the number of PT referrals for OA in 2020 compared to those placed in 2018 or 2019.

The comparison of the PT referral count data in 2019 and 2020 with month-to-month trends is shown in Fig 2. Comparing the two years, a total of 1972 PT referrals were placed from January through November 2019, compared to only 1614 referrals placed from January to November 2020 ($p = 0.001$), as shown in Table 2. Our analysis revealed significant month-to-month differences, with substantially fewer PT referrals for OA placed in April, May, and August 2020, all with adjusted $p$-values = 0.001. Interestingly, these dates fall immediately after the initial stay-at-home order issued by the Governor of the State of California on March 19, 2020 [8] and the decision made by the Health Officer of Sacramento County to continue this order, issued on July 14, 2020 [9].

Fig 3 documents the number of new COVID-19 cases diagnosed per day and per month in Sacramento County, California between February and November 2020. An initial surge in

**Table 1. 2018 referral data compared to 2019 and 2020.**

| Quarter | P-value | P-value |
|---|---|---|
| | 2018 vs 2019 | 2018 vs 2020 |
| Qtr1 | 0.33 | <0.001 |
| Qtr2 | 0.015 | <0.001 |
| Qtr3 | 0.001 | 0.02 |
| Qtr4 | 0.21 | <0.001 |

Displays p-values that document differences in referrals for PT in 2018 compared to those in 2019 and 2020. 2018 compared to 2019 in quarters 2 and 3 had significant differences, with 2019 having more referrals than 2018. We found that the number of referrals in the first quarter of 2020 was similar to those made in all quarters of 2019. Thus, 2019 was used as the control year for subsequent analysis. Overall, the findings revealed statistically significant reductions in the number of referrals to PT made in 2020 compared to 2018. Data were analyzed using analyzed via the Poisson Means Test.

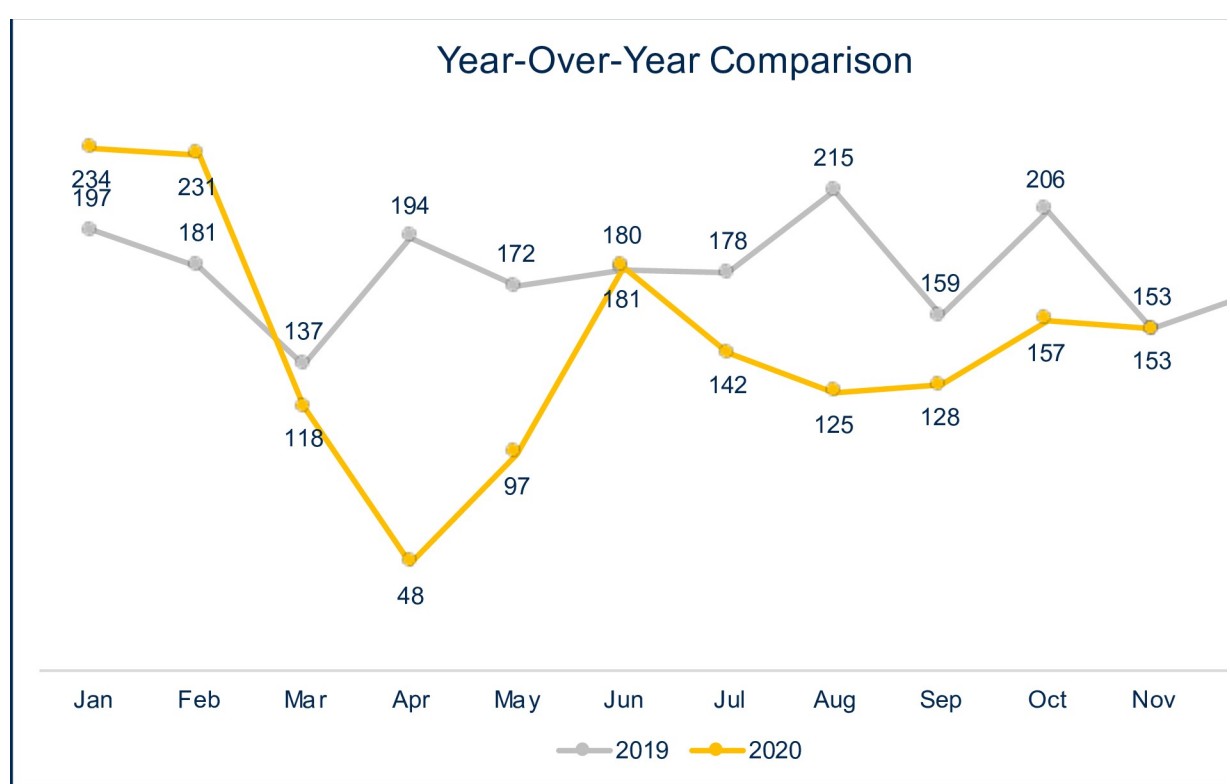

**Fig 2. Referrals for PT per month in 2019 *versus* 2020.** The numbers of PT referrals per month for the year 2019 are linked by the gray line. The numbers of PT referrals per month in 2020 are linked with the yellow line. Two major drops in referrals were seen in 2020.

cases was reported in March through April 2020 followed by a second larger surge that began in June 2020 and persisted through the end of the calendar year. By contrast, our findings revealed a reduction in the number of PT referrals beginning in March 2020 (Fig 2) that

**Table 2. Referral data in 2019 and 2020 by month.**

| Month | 2019 Referrals | 2020 Referrals | *P*-value | Adjusted *P*-value |
|---|---|---|---|---|
| Jan | 197 | 234 | 0.08 | 0.30 |
| Feb | 181 | 231 | 0.01 | 0.09 |
| Mar | 137 | 118 | 0.23 | 0.55 |
| Apr | 194 | 48 | <0.001 | 0.001 |
| May | 172 | 97 | <0.001 | 0.001 |
| June | 180 | 181 | 0.96 | 1.00 |
| July | 178 | 142 | 0.04 | 0.24 |
| Aug | 215 | 125 | <0.001 | 0.001 |
| Sept | 159 | 128 | 0.07 | 0.30 |
| Oct | 206 | 157 | 0.01 | 0.08 |
| Nov | 153 | 153 | 1.00 | 1.00 |
| Year Total | 1,972 | 1,614 | <0.001 | 0.001 |

Shown are referrals placed each month (January to November) in 2019 and 2020 with corresponding p-values. Data were analyzed using the Poisson Means Test. Adjusted p-values were generated by two-factor Poisson Regression with interactions between month and year with corresponding contrasts for year and months between years.

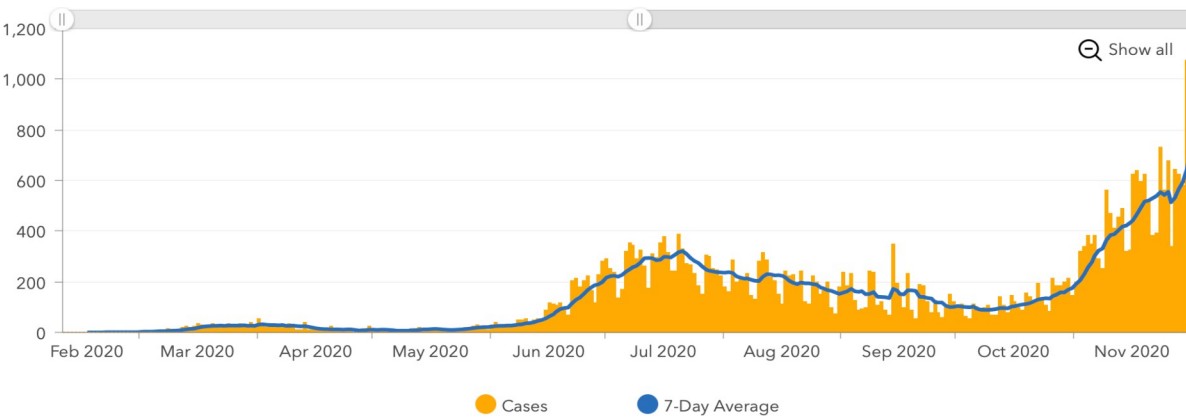

**Fig 3. Cases of COVID-19 in Sacramento County, California.** These cases are shown from February through November 2020. Included with the permission of the Sacramento County [7].

persisted and reached statistical significance in April 2020, both compared to 2019 data. We note that the State of California initiated Stage II of re-opening beginning in May 2020 [10]. We observed an overall increase in PT referrals for OA in June 2020 relative to May 2020. We also detected a significant drop in referrals in August 2020, shortly after the state-wide closures of bars and indoor business and gymnasiums on June 18 and July 2020, respectively [11]. Collectively, the findings presented in Table 2 and Fig 3 suggest that the number of PT referrals placed in 2020 fluctuated in a pattern that corresponded with state-mandated closures. Of interest, the pattern of referrals does not reflect the surges in COVID-19 cases reported in Sacramento County.

## Discussion

Collectively, our data indicate a statistically significant drop in PT referrals placed in 2020 compared to 2019 for patients diagnosed with OA. We identified two specific downslopes in referral trends beginning in March and July 2020, respectively. We note with interest that these decreases occurred just after the initial California state-mandated closures came into effect. This is somewhat surprising, as one might anticipate that the cancellation of elective surgical procedures necessitated by the pandemic would result in greater use of PT for managing patients with symptomatic OA. Of note, the UC Davis Ambulatory Care Center, which offers both primary care services and specialty clinics including PT, remained open and operating at normal capacity during pandemic.

While patient access to healthcare resources may have been curtailed during the COVID-19 pandemic, many potentially important factors might have contributed to the observed reduction in PT referrals. First and foremost, we must consider the possibility that there was an overall reduction in the number of patients seeking care for OA at our facilities during this period. Many patients lost access to transportation during the pandemic, and others may have chosen to avoid the risks associated with traveling to appointments. Similarly, health care providers may have deferred placing referrals for specialty services such as PT due to anticipated delays and/or difficulties in obtaining care during this time. These possibilities might be explored further, as they highlight the ongoing needs of the elderly and underserved.

Our study also highlights other factors that remain to be addressed. While many patients did not obtain health care during the COVID-19 pandemic, this was of particular concern for individuals belonging to ethnic minority groups and those residing in socio-economically

disadvantaged neighborhoods. Not only were these individuals at higher risk for increased severity and mortality associated with COVID-19 [12], OA is also more prevalent among ethnic minorities, including Hispanics and non-Hispanic Blacks [2]. It would be important to determine if reduced rates of PT referrals led to even greater health disparities. It might also be helpful to determine any differences in the type of health insurance (i.e., private payer, government, or no insurance) correspond with a specific pattern of PT referrals counts for patients with OA during this time period.

Among the limitations of our study, our findings are from a single-center retrospective study with a limited number of patients. As noted above, we recognize that many factors can contribute to the reduction in PT referrals, including those not directly associated with the COVID-19 pandemic. Our data focused on group rates and did not feature data collected at the individual patient level. Thus, longitudinal and logistic regression data analysis cannot be performed. Similarly, we did not collect suitable demographic data and thus we cannot perform a secondary analysis based on age, education, gender, inferred socioeconomic status, or type of insurance.

Likewise, while our data indicate that fewer PT referrals were placed overall during 2020 compared to 2019, it would be helpful to compare these findings to the number of overall patient visits to primary care centers during this time. This would help us to understand whether the reductions in PT referrals were due to an overall reduced number of patient visits as opposed to a specific reduction in the number of referrals per patient visit.

Furthermore, despite the ongoing pandemic, the sample size and duration of the study cannot be expanded given limited financial resources and time. This factor also limited the original study design, which does not include the clinical aspect of patient or physician opinions. Patient and physician surveys would be a helpful clinical aspect in identifying barriers for the reduced physical therapy referrals but were not incorporated into the study design. For future directions, this would be helpful in understanding the causes for the reduction in PT referrals for OA.

Finally, the data collected focused on referrals placed but did not differentiate between those that were pending *versus* those that were completed. Additional information might be gathered to address this point. This information may provide a stronger measure of overall access to PT during the pandemic.

## Conclusion

Collectively, our data indicate that significantly fewer PT referrals for patients with OA were placed in 2020 compared to 2019. These findings merit ongoing attention from both health systems and local public health officials. As but one suggestion, PT programs might be tailored so that they can be provided to patients online, thereby encouraging ongoing participation via socially distanced healthcare. We note the study by Miller et al [13] who found that virtual PT was both feasible and acceptable, with 94% of patients reporting satisfaction with this modality. An exploration of how this option might be implemented on a large scale might help to limit delays and address ongoing deficiencies in patient care during the pandemic.

## Supporting information

**S1 File. OA ICD-10 codes.** 206 applicable ICD-10 diagnosis codes for OA (see Supplemental Data 1) that were linked to PT referrals in our data set.
(DOCX)

**S2 File. Raw PT referral data.** Excel file with PT referrals (de-identified and used for analysis).
(XLSX)

**S3 File. Sacramento County permission.** Permission to Use Fig 3 from the Sacramento County who publishes the COVID cases dashboard online.
(DOCX)

## Author Contributions

**Conceptualization:** Manmeet Kaur, Barton L. Wise.

**Data curation:** Manmeet Kaur, Daniel Black.

**Formal analysis:** Manmeet Kaur, Daniel Black, Jeffrey Fine.

**Investigation:** Manmeet Kaur, Daniel Black.

**Methodology:** Manmeet Kaur.

**Resources:** Manmeet Kaur.

**Software:** Daniel Black, Jeffrey Fine.

**Supervision:** Barton L. Wise.

**Validation:** Manmeet Kaur, Daniel Black.

**Visualization:** Manmeet Kaur, Daniel Black.

**Writing – original draft:** Manmeet Kaur.

**Writing – review & editing:** Manmeet Kaur, Jeffrey Fine, Barton L. Wise.

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
