## [Decision Letter · Decision Letter 0]

9 Aug 2021

PONE-D-21-22615

Referrals for Physical Therapy for Osteoarthritis during the COVID-19 Pandemic:  a retrospective analysis

PLOS ONE

Dear Dr. Kaur,

Thank you for submitting your manuscript to PLOS ONE. After careful consideration, we feel that it has merit but does not fully meet PLOS ONE’s publication criteria as it currently stands. Therefore, we invite you to submit a revised version of the manuscript that addresses the points raised during the review process.

We look forward to receiving your revised manuscript.

Kind regards,

Sinan Kardeş, M.D.

Academic Editor

PLOS ONE

Journal Requirements:

Reviewers' comments:

Reviewer's Responses to Questions

**Comments to the Author**

1. Is the manuscript technically sound, and do the data support the conclusions?

Reviewer #1: Yes

Reviewer #2: No

2. Has the statistical analysis been performed appropriately and rigorously? 

Reviewer #1: Yes

Reviewer #2: No

3. Have the authors made all data underlying the findings in their manuscript fully available?

Reviewer #1: Yes

Reviewer #2: No

4. Is the manuscript presented in an intelligible fashion and written in standard English?

Reviewer #1: Yes

Reviewer #2: Yes

5. Review Comments to the Author

Reviewer #1: To Authors,

You conducted a well-written study assessing the effect of COVID-19 pandemic on the physical therapy referrals for patients with osteoarthritis. The study is interesting. However, the lack of clinical perspective makes the study incomplete.

- Please specify the institution where the study was conducted only in the method section.

- Please remove figure 2 and figure 4 and create the tables instead of them.

- Could you improve the quality of figures?

- Please avoid repeating the aim of the study.

- The underlying reasons for this reduction may be varied, as you mentioned in the introduction and discussion sections. Why did you not assess physicians and/or patients' opinion regarding PT referrals (for ex., with questionnaires) during pandemic?

- Could you expand the study limitations in the discussion section?

Reviewer #2: This paper was generally well-written and focuses on physical therapy referrals during the COVID-19 Pandemic.

My major issue with this paper concerns the statistical analyses and conclusions drawn. The authors frequently use language (e.g., "influence") that indicates a casual relationship between COVID-19 and a decrease in physical therapy referrals; this is not correct. The authors did not run appropriate statistical analyses to determine a casual effect between the two. Unless a longitudinal, within-person design is used, causation can not be inferred, only correlation. This is especially true given no covariates (e.g., age, sex, motivation levels) were accounted for. The authors need to modify their language throughout to reflect this.

If the authors do not have data to run a longitudinal design, I would recommend a logistic regression (comparing years) to determine whether the odds of receiving physical therapy referrals decreased from year-to-year. Although this would still not give indication of causation, it would provide a more robust assessment of the authors' hypotheses.

6. PLOS authors have the option to publish the peer review history of their article (what does this mean?). If published, this will include your full peer review and any attached files.

Reviewer #1: **Yes: **Neslihan Gokcen

Reviewer #2: No

---

## [Author Response · Author response to Decision Letter 0]

23 Sep 2021

Thank you for the reviewers’ valued feedback on our manuscript entitled “Referrals for Physical Therapy for Osteoarthritis during the COVID-19 Pandemic: a retrospective analysis” (ID: PONE-D-21-22615). According to the reviewers’ suggestions, we have made careful revisions of the original manuscript. Please find the response to the comments below each point presented by the reviewers, in addition to updates made accordingly to the manuscript. Thank you for your time and considering our edits. 

Replies to the reviewers’ comments: 

Reviewer #1: To Authors,

- Please specify the institution where the study was conducted only in the method section. 

Response: Manuscript updated to address this point

- Please remove figure 2 and figure 4 and create the tables instead of them.

Response: We do not entirely understand the reviewer’s comment, as both figures are in fact tables. However, to help address this point, the titles of Figures’ 2 and 4 were changed to Table 1 and 2, respectively, to better represent content. And Figure 1, 3, 5 (which were graphs), were appropriately re-numbered as Figure 1, 2, and 3, respectively.

- Could you improve the quality of figures? 

Response: Uploaded with improved quality.

- Please avoid repeating the aim of the study. 

Response: Updated the manuscript to address this.

- The underlying reasons for this reduction may be varied, as you mentioned in the introduction and discussion sections. Why did you not assess physicians and/or patients' opinion regarding PT referrals (for ex., with questionnaires) during pandemic? 

Response: This was a limitation of the study given time constraint and original study design. The clinical aspect would be helpful in identifying barriers from both the patient and physician side, but was not incorporated into the original IRB approval and project design. The goal was to identify whether this gap in healthcare for physical therapy (PT) for osteoarthritis (OA) existed during the pandemic. For future directions, this would be helpful in understanding potential causes for the reduction in PT referrals for OA.

- Could you expand the study limitations in the discussion section?

Response: Addressed the limitations in the discussion section of the manuscript, please find an overview listed here as well. 

"Among the limitations of our study, our findings are from a single-center retrospective study with a limited number of patients. As noted above, we recognize that many factors can contribute to the reduction in PT referrals, including those not directly associated with the COVID-19 pandemic. Our data focused on group rates and did not feature data collected at the individual patient level. Thus, longitudinal and logistic regression data analysis cannot be performed. Similarly, we did not collect suitable demographic data and thus we cannot perform a secondary analysis based on age, education, gender, inferred socioeconomic status, or type of insurance. 

Likewise, while our data indicate that fewer PT referrals were placed overall during 2020 compared to 2019, it would be helpful to compare these findings to the number of overall patient visits to primary care centers during this time. This would help us to understand whether the reductions in PT referrals were due to an overall reduced number of patient visits as opposed to a specific reduction in the number of referrals per patient visit. 

Furthermore, despite the ongoing pandemic, the sample size and duration of the study cannot be expanded given limited financial resources and time. This factor also limited the original study design, which does not include the clinical aspect of patient or physician opinions. Patient and physician surveys would be a helpful clinical aspect in identifying barriers for the reduced physical therapy referrals but were not incorporated into the study design. For future directions, this would be helpful for understanding the causes for the reduction in PT referrals for OA.

Finally, the data collected focused on referrals placed but did not differentiate between those that were pending versus those that were completed. Additional information might be gathered to address this point. This information may provide a stronger measure of overall access to PT during the pandemic."

Reviewer #2: This paper was generally well-written and focuses on physical therapy referrals during the COVID-19 Pandemic.

My major issue with this paper concerns the statistical analyses and conclusions drawn. The authors frequently use language (e.g., "influence") that indicates a casual relationship between COVID-19 and a decrease in physical therapy referrals; this is not correct. The authors did not run appropriate statistical analyses to determine a casual effect between the two. Unless a longitudinal, within-person design is used, causation can not be inferred, only correlation. This is especially true given no covariates (e.g., age, sex, motivation levels) were accounted for. The authors need to modify their language throughout to reflect this.

If the authors do not have data to run a longitudinal design, I would recommend a logistic regression (comparing years) to determine whether the odds of receiving physical therapy referrals decreased from year-to-year. Although this would still not give indication of causation, it would provide a more robust assessment of the authors' hypotheses.

Response: Unfortunately, there is no person-level data, so a longitudinal and logistic regression cannot be done. However, a two-factor Poisson Regression with an interaction between month and year with corresponding contrasts for year and month between 2019 and 2020 was performed (Table 1) for a more robust statistical analysis. Language has been updated in the Results/Discussion section to reflect this new analysis. This was also emphasized in the limitations section of the discussion.

The language was updated in manuscript to reduce any causal implications, and instead emphasize correlation. 

End of Reply--------------------------------------------- 

Once again, thank you very much for your constructive feedback and suggestions. Please let me know if there are any additional questions or concerns. 

Manmeet Kaur, MD (Corresponding author)

mmkaur@ucdavis.edu

---

## [Decision Letter · Decision Letter 1]

25 Oct 2021

Referrals for Physical Therapy for Osteoarthritis during the COVID-19 Pandemic:  a retrospective analysis

PONE-D-21-22615R1

Dear Dr. Kaur,

We’re pleased to inform you that your manuscript has been judged scientifically suitable for publication and will be formally accepted for publication once it meets all outstanding technical requirements.

Kind regards,

Sinan Kardeş, M.D.

Academic Editor

PLOS ONE

Reviewers' comments:

Reviewer's Responses to Questions

**Comments to the Author**

1. If the authors have adequately addressed your comments raised in a previous round of review and you feel that this manuscript is now acceptable for publication, you may indicate that here to bypass the “Comments to the Author” section, enter your conflict of interest statement in the “Confidential to Editor” section, and submit your "Accept" recommendation.

Reviewer #1: All comments have been addressed

Reviewer #2: All comments have been addressed

2. Is the manuscript technically sound, and do the data support the conclusions?

Reviewer #1: Partly

Reviewer #2: Yes

3. Has the statistical analysis been performed appropriately and rigorously? 

Reviewer #1: Yes

Reviewer #2: Yes

4. Have the authors made all data underlying the findings in their manuscript fully available?

Reviewer #1: Yes

Reviewer #2: Yes

5. Is the manuscript presented in an intelligible fashion and written in standard English?

Reviewer #1: Yes

Reviewer #2: Yes

6. Review Comments to the Author

Reviewer #1: (No Response)

Reviewer #2: All of my comments have been adequately addressed. The paper now takes a more robust statistical approach and is improved from the previous version.

7. PLOS authors have the option to publish the peer review history of their article (what does this mean?). If published, this will include your full peer review and any attached files.

Reviewer #1: **Yes: **Neslihan Gokcen

Reviewer #2: No

---

## [Editor Report · Acceptance letter]

29 Oct 2021

PONE-D-21-22615R1 

Referrals for Physical Therapy for Osteoarthritis during the COVID-19 Pandemic:  a retrospective analysis 

Dear Dr. Kaur:

I'm pleased to inform you that your manuscript has been deemed suitable for publication in PLOS ONE. Congratulations! Your manuscript is now with our production department. 

Kind regards, 

on behalf of

Dr. Sinan Kardeş 

Academic Editor

PLOS ONE